# EFFICIENT MULTI-FIDELITY NAS WITH ZERO-COST PROXY-GUIDED LOCAL SEARCH

## ABSTRACT

Using zero-cost (ZC) metrics as proxies for network performance is currently trendy in Neural Architecture Search (NAS) because the low computing cost of these metrics allows search algorithms to thoroughly explore the architecture search space. Nevertheless, recent studies indicate that relying exclusively on ZC proxies appears to be less effective than using traditional training-based metrics, such as validation accuracy, in seeking high-performance networks. Training-based metrics are preferred as the main search objective to guide search algorithms to approach truly good architectures while ZC proxies could be used as low-cost surrogates to accelerate the search process. ZC proxies with high rank-correlations to network test accuracy are supposed to bring better search results than metrics with lower correlations. In this study, we investigate the effectiveness of ZC proxies in NAS by taking a deeper look into their fitness landscapes rather than focusing only on rank correlations. We construct fitness landscapes of ZC proxy-based local searches by utilizing the Local Optima Network (LON), which is a powerful visualization tool to analyze combinatorial optimization problems. Our findings exhibit that a high correlation does not guarantee finding high-performance architectures, and ZC proxies with low correlations could still be better in certain situations. Our results further consolidate the suggestion of favoring training-based metrics over ZC proxies as the search objective. Although we could figure out the architectures having the optimal ZC proxy scores, their true performance is often poor. We then utilize insights from our landscape analysis to propose **M**ulti-**F**idelity **N**eural **A**rchitecture **S**earch (MF-NAS), which is a novel framework that makes use of the efficiency of ZC proxies and the efficacy of training-based metrics. Experimental results on a wide range of NAS benchmarks, i.e., NAS-Bench-101, NAS-Bench-201, and NAS-Bench-ASR, demonstrate the superiority of our proposed approach to state-of-the-art NAS methods under a strict budget.

## 1 INTRODUCTION

Neural architecture search (NAS) has demonstrated its potential in designing powerful neural networks when network models found by NAS might surpass the manually-designed ones (Zoph & Le, 2017; Tan & Le, 2019; Chebykin et al., 2022). In NAS, the *performance evaluation strategy*, which estimates the true performance of architectures (e.g., test accuracy), plays a vital role in guiding the search algorithms toward high-quality architectures. Many metrics have been utilized as proxies for test performance and they could be divided into two categories: *training-based* (Xie & Yuille, 2017; Ru et al., 2021) and *training-free* (Tanaka et al., 2020; Mellor et al., 2021; Abdelfattah et al., 2021).

One of the commonly-used training-based metrics is validation accuracy (Xie & Yuille, 2017; Zoph & Le, 2017; White et al., 2021a). Although the computing cost of training-based metrics are high (due to expensive network training to obtain proper network weights before evaluating on the validation set), these metrics are reliable and highly correlated to the test accuracy. Some techniques have been proposed to deal with the high cost of training-based metrics such as learning curve extrapolation (Yan et al., 2021) or using *training-free* metrics (Tanaka et al., 2020; Mellor et al., 2021; Abdelfattah et al., 2021). Training-free metrics (also known as *zero-cost* proxies in the literature) are performance indicators that can be computed with a single pass through the network (i.e., without a typical training procedure of many iterations); the costs for obtaining their values are thus trivial (Mellor et al., 2021; Krishnakumar et al., 2022). The number of parameters and the

floating-point operations per second (FLOPS) are also considered as the baselines among zero-cost (ZC) proxies (Ning et al., 2021; White et al., 2022; Krishnakumar et al., 2022). In addition to the number of parameters and FLOPs, many ZC proxies have been proposed and have shown promising Spearman's rank correlations with network accuracy such as Synaptic Flow (Tanaka et al., 2020) or NWOT (Mellor et al., 2021). However, recent studies indicate that using *solely* ZC proxies as the search objective seems to be ineffective in finding truly good architectures. Instead, ZC proxies are suggested to be used to accelerate the NAS process, and training-based metrics should still be the main objective to guide the search algorithms (Abdelfattah et al., 2021; White et al., 2022; Krishnakumar et al., 2022).

In this study, we first investigate the capability of ZC proxies to search for top-performing architectures. While previous studies mainly focus on rank correlations between ZC proxies and test performance, we put more attention on the *fitness landscapes* of these training-free metrics. We construct fitness landscapes by utilizing Local Optima Network (LON) (Ochoa et al., 2008), which is a visualization tool for the analysis of combinatorial optimization problems (Baioletti et al., 2019; Zou et al., 2022). Based on insights derived from our analysis, we then propose a novel multi-fidelity NAS framework that makes use of the strengths of both training-free and training-based metrics. Our contributions are summarized as follows:

- This study is the pioneer that evaluates the effectiveness of ZC proxies by visualizing and analyzing their fitness landscapes via Local Optima Networks (LONs). Our analysis highlights the failure of ZC proxies in obtaining high-quality architectures because the majority of the networks that attain optimal ZC proxy scores actually yield poor test performance. High correlations of ZC proxies with test performance might not guarantee that ZC proxy-based search would discover powerful architectures. Nevertheless, the insights from LONs demonstrate the suitability of the fitness landscapes of ZC proxies for a typical local search with a simple escape operator to straightforwardly approach the regions containing promising networks.

- We introduce an efficient two-stage Multi-Fidelity NAS (**MF-NAS**) framework. The first stage involves exploring the search space using a local search algorithm guided by a ZC proxy. In the second stage, the top-scored candidate architectures obtained in the first stage are evaluated using a training-based metric in an efficient manner via the Successive Halving procedure. The network with the highest validation accuracy is selected as the final NAS result, yielding both the architecture together with its network weights.

- We perform rigorous experiments to compare **MF-NAS** with many state-of-the-art NAS methods in NAS-Bench-101/201/ASR search spaces.

## 2 RELATED WORK

**Zero-Cost Proxy Analysis** Some ZC proxies have been found to exhibit biases for larger-sized networks (Ning et al., 2021), narrower networks, or networks with wider channels (Chen et al., 2021a). Relying solely on these metrics could divert the search algorithm from its primary goal of discovering high-performance architectures. Several studies evaluated the effectiveness of ZC metrics in finding top candidate architectures through their rank correlations with the test accuracy and suggested using these training-free metrics as "low-cost surrogates" to accelerate the search process rather than the main objective for guiding the search algorithm (Abdelfattah et al., 2021; White et al., 2022; Krishnakumar et al., 2022).

**NAS Landscapes Analysis** Conducting fitness landscape analysis[1] is an intuitive approach to gain useful insights from NAS search spaces. White et al. (2021b) analyzed the loss landscape and suggested using local search as a strong baseline algorithm in NAS. They also indicated the search difficulty would reduce if the noise in the training pipeline could be decreased. Similar conclusions about local search were given in Ochoa & Veerapen (2022) and Thomson et al. (2023), where the

---

[1]A fitness landscape is composed of three components: (1) a set of all feasible solutions in the search space $S$, (2) a neighborhood function $N \colon S \to 2^S$ that assigns a set of neighboring solutions to every $s \in S$, and (3) a fitness function $f \colon S \to \mathbb{R}$ that maps solutions to their corresponding coordinates in the objective space. (Reidys & Stadler, 2001)

fitness landscapes were also visualized with local optima networks (LONs)[2], and local search was deemed preferable to other algorithms in NAS. However, Ochoa & Veerapen (2022) and Thomson et al. (2023) considered only the case when the validation accuracy after 200 training epochs was the search objective. To our knowledge, there is no study so far that analyzes the landscapes of ZC proxies as the objective and uses LONs for visualizing these training-free metrics' fitness landscapes.

**Successive Halving**  Successive Halving (SH) (Jamieson & Talwalkar, 2016) is a multi-fidelity mechanism that is often employed in NAS (Baker et al., 2018; Wang et al., 2021). Inputting a set of candidate architectures, SH first scores all of them using a training-based metric at a low fidelity (i.e., architectures are trained for just a few epochs and then evaluated for validation accuracy or training loss). Subsequently, the half of candidates having worse scores is discarded while the better half undergoes higher-fidelity evaluations (i.e., the survived networks are trained for more epochs). This process is iterated until there remains only one candidate or the allocated computing budget is exhausted. The core concept behind SH is that the most promising candidates receive progressively higher-fidelity evaluations. The last surviving candidates are deemed the best solutions, and they are evaluated with the highest fidelity. In our MF-NAS, we utilize SH as the selection algorithm in the second stage to return the best candidate architecture.

## 3   FAILURE OF ZC PROXIES IN SEEKING TOP-PERFORMING ARCHITECTURES

We explore the NAS-Bench-201 search space (Dong & Yang, 2020; Dong et al., 2022) using the **best-improvement hill climbing algorithm** (i.e., at each iteration, moving to the best solution in the 1-opt neighborhood of the current solution if the new solution is better) with 13 zero-cost NAS proxies in *NAS-Bench-Suite-Zero* (Krishnakumar et al., 2022) as the search objective: **EPE-NAS** (Lopes et al., 2021), **Fisher** (Turner et al., 2020), **Grad-norm** (Abdelfattah et al., 2021), **Grasp** (Wang et al., 2020), **L2-norm** (Abdelfattah et al., 2021), **Jacov** (Mellor et al., 2021), **NWOT** (Mellor et al., 2021), **Plain** (Abdelfattah et al., 2021), **Synflow** (Tanaka et al., 2020), **Snip** (Lee et al., 2019), **Zen** (Lin et al., 2021), **FLOPS**, and **Params** (i.e., the number of parameters). We also experiment with three training-based metrics (i.e., validation accuracy, validation loss, and training loss) for comparisons. Since early stopping is typically used in real-world NAS to avoid having to train each candidate architecture for hundreds of epochs, we here also query the benchmark at the $12^{th}$ epoch for all training-based metric evaluations.

For each metric, we perform 15,625 attempts of hill climbing corresponding to using all 15,625 architectures in the search space as the initial solutions. Whenever the algorithm cannot find any improvement over an architecture $x$ within its 1-opt neighborhood, we would consider $x$ a local optimum in the search space. We further check whether there exist transitions between these optima by performing an **escape** operator. If the hill climbing is stuck at a local optimum $x$, we randomly sample another solution $x'$ in the 2-opt neighborhood of $x$ and then execute the hill climbing process again using $x'$ as the initial solution. This escape procedure is performed on all solutions in the 2-opt neighborhood to completely evaluate the possibility of reaching the global optimum from an arbitrary initial solution. We employ the *Monotonic Local Optima Network* (MLON) (Ochoa et al., 2017), which is a type of LON that only exhibits the connections from low-value nodes to high-value nodes, to visualize the exploration results. The nodes symbolize the optima, and the edges represent the transitions between optima using the escape operator. The size and color of each node are proportional to the total number of edges coming towards that node and its metric value, respectively. The width of an edge $e_{ij}$ is proportional to the number of architectures in the 2-opt neighborhood of the solution at node $i$ that can lead the hill climbing to the solution at node $j$.

Fig. 1 and Fig. 2a present that the landscapes of **FLOPS** and **Params** are uni-modal with respect to all three datasets, as well as the **Synflow** metric for CIFAR-100 and ImageNet16-120. These LONs indicate that when we use these metrics as the search objective, given enough runtime, hill climbing always reaches the architecture having the maximum number of parameters or the greatest FLOPs in NAS-Bench-201 without invoking the escape operator. In contrast, the LONs of other metrics exhibit that their fitness landscapes are multi-modal. Some landscapes are sparse, as seen

---

[2]Local optima network (LON) is defined as a weighted-oriented graph used to structure the fitness landscape. Nodes in an LON represent local optima, and the edge $e_{ij}$ from node $i$ to node $j$ presents that we can move from the optimum $i$ to the optimum $j$ via a pre-defined escape operator. (Ochoa et al., 2008)

in **NWOT** and **GRASP**, while others are dense such as **Jacov**, **EPE-NAS**, and **Plain**. The MLONs in Fig. 1 further present that there always exists direct (or indirect) connections from a low-value optimum to higher-value optima. Therefore, when we conduct NAS with one of these ZC proxies as the search objective, we are guaranteed to always figure out the optimal architecture with respect to that metric in NAS-Bench-201 by using a local search algorithm with a simple escape operator despite the landscapes being multi-modal.

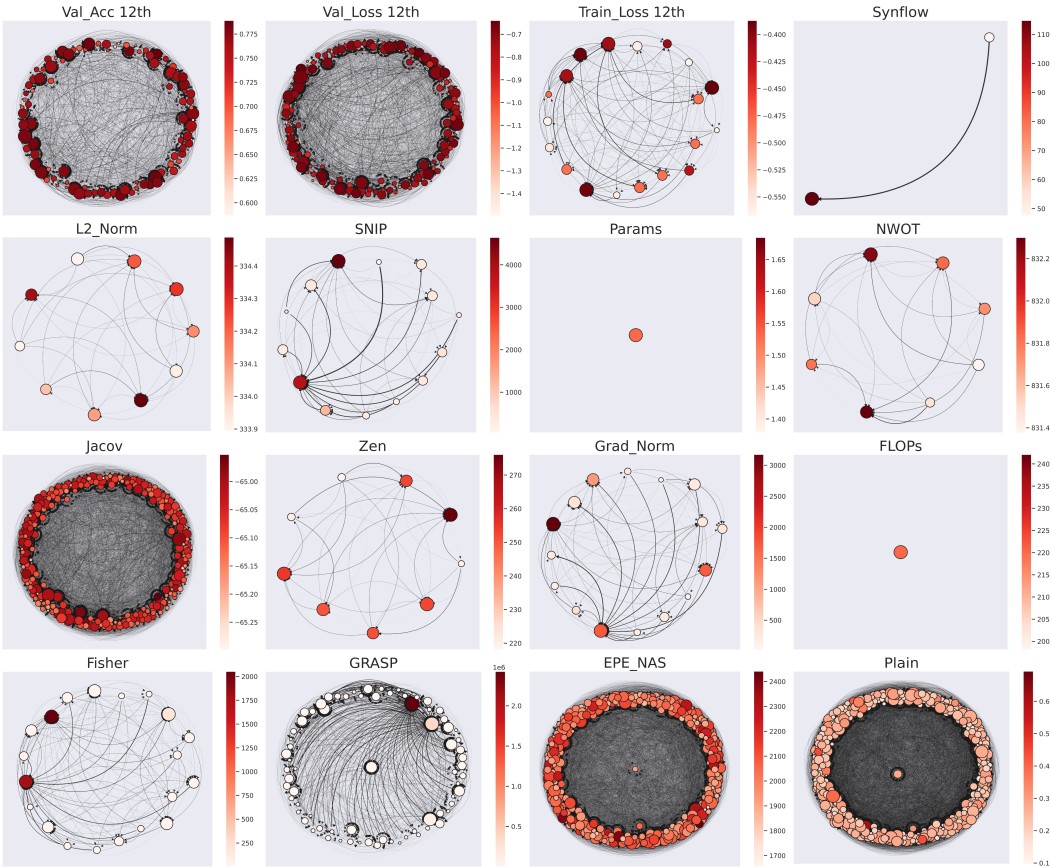

Figure 1: MLONs of the best-improvement hill climbing algorithm with escape for CIFAR-10 on NAS-Bench-201. The MLONs for CIFAR-100 and ImageNet16-120 are in Appendix A.1.1.

Fig. 2b exhibits a notable difference in the test performance between optimal solutions, especially for the ones found by ZC proxies. For training-based metrics, the global optima on the training loss landscapes are the best at the test accuracy. Together with the insight from MLONs, this finding helps clarify the results in Ru et al. (2021) that the training loss outperforms other training-based metrics on NAS-Bench-201. On the other hand, most architectures at global optima on the landscapes of ZC proxies yield poor test accuracy and *are worse than the ones on the training-based metrics' landscapes*. High rank correlation to the test accuracy is not the main factor in ensuring the finding of top-performing architectures. For instance, the **NWOT** metric has a Spearman's correlation of approximately 0.8, which is the highest correlation among ZC proxies, but the optimal solutions found by **NWOT**-guided local search are not the best ones in terms of test performance compared to those of other ZC proxies (see Fig. 2b and Fig. 2c). Some ZC proxies such as **Snip** and **Grad-norm** have a correlation of approximately 0.6 but their optimal solutions are worse than the ones found by **Zen**, which only has a correlation of approximately 0.3. We also found that *none of the solutions at the global optima* achieve the optimal test accuracy (see Fig. 2b).

We additionally discover that the regions of top-1% networks in terms of ZC proxy scores contain few top-performing architectures (see Fig. 3) and almost all the global optima on ZC proxies' landscapes are not in the top-1% most-accurate architectures. These findings indicate that even if we could approach the regions of top-1% in terms of ZC proxy scores, obtaining the architectures

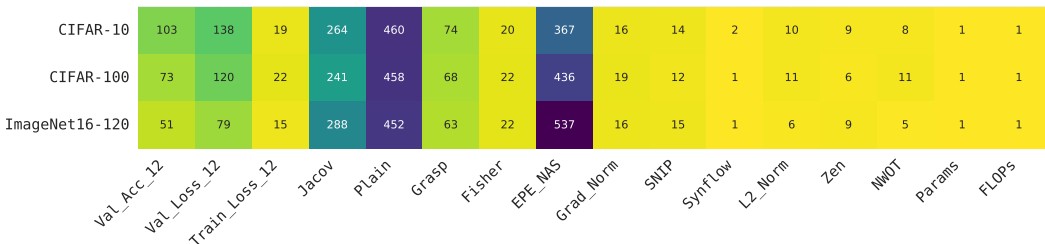

(a) The number of optima on the fitness landscapes of each metric.

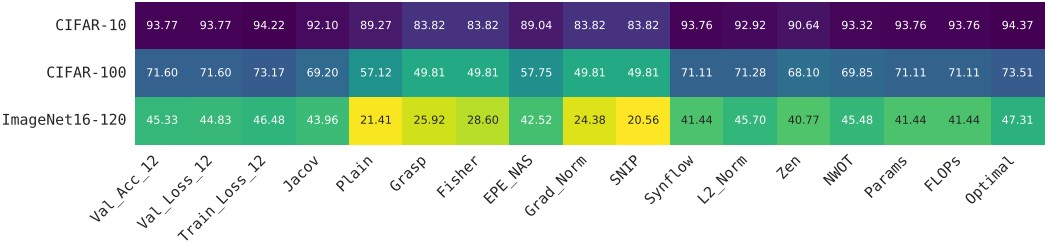

(b) The test accuracy of architectures at the global optima on the fitness landscapes of each metric.

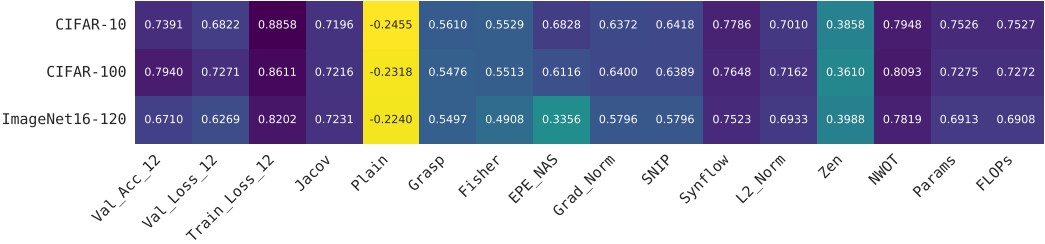

(c) The Spearman's rank correlation between the test accuracy values and the scores of each metric.

Figure 2: The results of exploring different fitness landscapes on NAS-Bench-201.

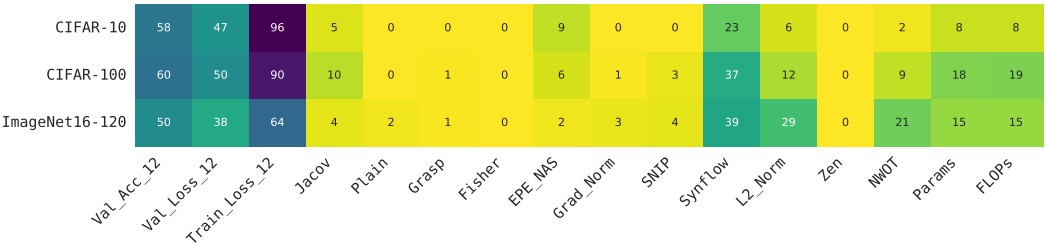

Figure 3: The number of top-1% most-accurate architectures (i.e., 156 best networks) within the top-1% architectures returned by each metric.

that truly have the highest test accuracy is difficult. NAS algorithms using these training-free metrics as the search objective might overlook the truly good architectures and return the poorer ones. We then perform experiments with **first-improvement hill climbing** (i.e., at each iteration, moving randomly to the first better solution in the 1-opt neighborhood of the current solution) on NAS-Bench-201. The effect of ZC proxies on the search behavior of the hill climbing algorithm with escape is demonstrated in Fig. 4. We can see that the search processes using **Grasp**, **Grad-norm**, or **Snip** as the objective function have found top-performing architectures quite early but then overlook them and finally converge at the ones having poorer performance. Despite incurring much more computation cost, using training-based metrics is more effective than using training-free metrics in seeking top-performing architectures.

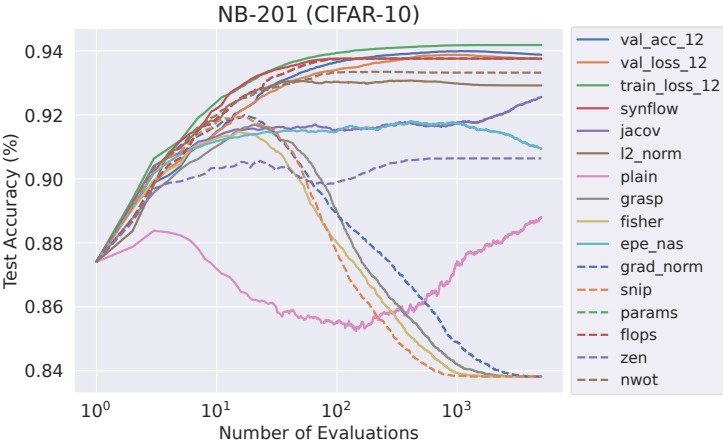

Figure 4: The average performance trends of the first-improvement hill climbing algorithm with escape regarding 3 training-based metrics and 13 ZC proxies over 5,000 evaluations on CIFAR-10. We repeat the search process 500 times for each metric. The results for CIFAR-100 and ImageNet16-120 can be found in Appendix A.1.1.

# 4 MULTI-FIDELITY NEURAL ARCHITECTURE SEARCH (MF-NAS)

We have two observations about the two categories of NAS metrics, i.e., training-based metrics and ZC proxies, via our landscape analysis in Section 3: **(1)** The characteristics of performance metrics' landscapes enable local search-based algorithms to reach the regions that potentially contain high-performing architectures. **(2)** Finding top-performing architectures by using *solely* ZC proxies is hard. Meanwhile, it is more straightforward if we utilize training-based metrics.

We propose to take advantage of ZC proxies with their low computation costs to guide the search algorithm toward the region of promising architectures, and then employ training-based metrics such as validation accuracy to seek for the most-accurate architecture in this region. We introduce the Multi-Fidelity Neural Architecture Search framework (MF-NAS), in which the search process is divided into two separate stages (see Appendix A.2 for pseudocode). The first stage is to explore the search space by the **first-improvement hill climbing algorithm with escape** that uses a ZC proxy $Z$ as the search objective (i.e., the low-fidelity stage). The purpose of this stage is to guide the algorithm toward search space regions containing promising architectures *with a trivial cost*. The budget of the local search stage is $M$ architecture evaluations, i.e., computing ZC proxy scores for $M$ candidate architectures. We log the ZC proxy values of all $M$ architectures found during the local search trajectory and then select the top-$k$ architectures having the highest scores to form a candidate pool $P$ at the end. This first stage is similar to the *Zero-Cost Warmup* procedure in Abdelfattah et al. (2021), but we note that they use Random Search to sample $M$ random architectures while we use a local search algorithm instead.

The architectures in the candidate pool $P$ are then evaluated using a training-based metric $T$ to identify the best one at the second stage (i.e., the high-fidelity stage). Specifically, we deploy the Successive Halving mechanism to reduce the expensive cost of network training. At the beginning of the second stage, all architectures in the candidate pool $P$ are evaluated for their training-based metric values after undergoing a few training epochs. The better half of the candidate pool with the highest training-based metric scores are kept while the worse half are eliminated from $P$. The selected architectures are then trained for more epochs (i.e., increasingly higher fidelity), and we repeat this evaluation-and-selection process on $P$ till the computing budget is exhausted or there remains only one architecture in $P$. In case the budget runs out before the pool $P$ is narrowed down to a single candidate, the architecture having the highest training-based metric value so far is considered the resulting solution found by the algorithm.

# 5 EXPERIMENTS

We evaluate MF-NAS on three NAS benchmarks: NAS-Bench-101 (Ying et al., 2019), NAS-Bench-201 (Dong & Yang, 2020; Dong et al., 2022), and NAS-Bench-ASR (Mehrotra et al., 2021). We also experiment with other NAS algorithms: Random search (RS), Local search (LS), Successive Halving (SH), Regularized evolution algorithm (REA) (Real et al., 2019), REA using the *Warmup* method (REA+W) (Abdelfattah et al., 2021), and FreeREA (Cavagnero et al., 2023). Hyperparameters for all algorithms are listed in Appendix A.3. Each algorithm is run 500 times independently as in Ying et al. (2019); Mellor et al. (2021); Dong et al. (2022). Comparisons with other state-of-the-art NAS algorithms can be found in Appendix A.4.

## 5.1 RESULTS ON NAS-BENCH-201

We first verify the efficacy of MF-NAS on NAS-Bench-201. We experiment with a total of 13 MF-NAS variants by trying out all ZC proxies considered in Section 3 as the ZC proxy $Z$. All variants employ validation accuracy as the training-based metric $T$ for the second stage. For the local search stage, we set the maximum number of evaluations $M$ to 2,000 and select out top-32 solutions (i.e., $k = 32$) that have the highest ZC proxy scores for the next stage. We set the total allowed computation budget $B$ following the experimental settings in Dong et al. (2022), i.e., 20,000 seconds for CIFAR-10, 40,000 seconds for CIFAR-100, and 120,000 seconds for ImageNet16-120. The settings for Successive Halving in MF-NAS are listed in Appendix A.3.

### 5.1.1 EFFECTIVENESS OF SUCCESSIVE HALVING IN SELECTING TOP-PERFORMING ARCHITECTURES

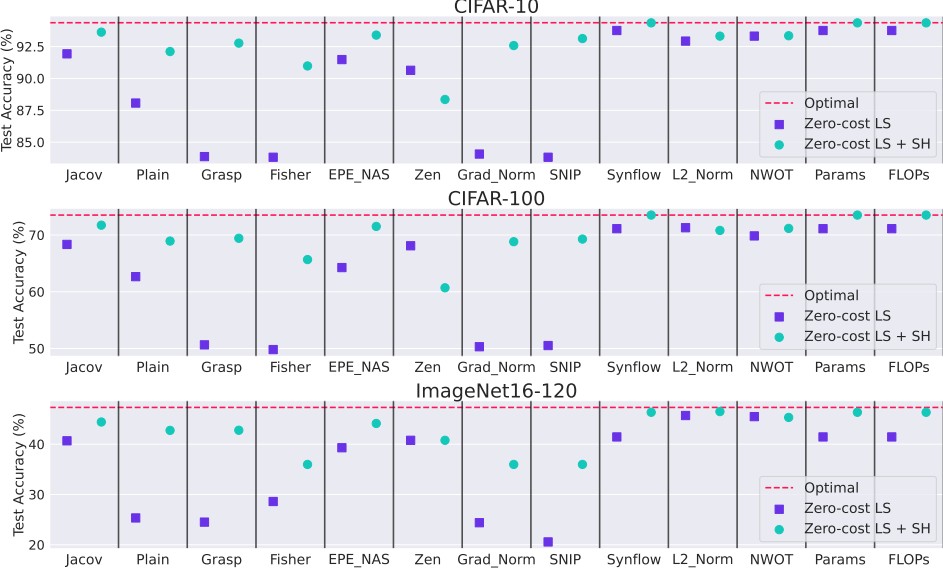

Figure 5: The average test accuracy performance of the best architectures returned at the end of the 1st stage (LS) and the 2nd stage (SH) of MF-NAS over 500 trials on NAS-Bench-201.

Fig. 5 exhibits the average test accuracy of the best architectures found at the two stages of MF-NAS. The performance gaps between the architectures having the best ZC proxy scores in the local search stage (i.e., the square markers) and the optimal test accuracy (i.e., the dashed lines) are substantial. However, the gaps are clearly reduced after the Successive Halving stage (i.e., the gap between the circle markers and the dashed lines). Fig. 5 also shows the large variance in the performance between architectures found by local search with different ZC proxies. In contrast, the two-stage MF-NAS exhibits impressive stability in the sense that all MF-NAS variants perform consistently well even though they utilize different ZC proxies for exploring the search space.

We consider the effectiveness of Successive Halving (SH) in selecting truly top-performing architectures by comparing the test accuracy between the architectures chosen by SH and the true best architectures from the input candidate pool (see Table 4, Appendix A.1.2). The majority of the architectures selected by SH are in the top three most-accurate architectures out of 32 input candidates. However, there are instances where SH returns poor candidates, such as in the case of the **Zen** metric on CIFAR-10. We then experiment with training loss as the training-based metric $T$ for the second stage of MF-NAS, and this change leads to an improvement in the quality of architecture selections for most cases (see Table 5). Training loss indeed exhibits a higher rank correlation with test accuracy compared to validation accuracy (see Fig. 2c).

### 5.1.2 EFFECTIVENESS OF LOCAL SEARCH IN EXPLORING THE SEARCH SPACE

We assess the impact of the first-improvement local search on the overall performance of the MF-NAS framework by comparing with the MF-NAS variants that use random search in the first stage. Using random search with ZC proxies to sample the search space is exactly the *Zero-Cost Warmup* method in Abdelfattah et al. (2021).

Table 1 exhibits that random search (RS) brings better results than local search (LS) in the case of **Plain**, **Fisher**, and **Zen** while local search is better for **Synflow**, **Params**, and **FLOPs**. The differences are because the top regions of the latter metrics consist of a great number of high-performance architectures while the former metrics do not have such networks in their top regions (see Fig. 3). LS is demonstrated as effective in approaching the top regions of ZC proxies but the non-existence of truly top-performing architectures in these regions makes its exploration less useful than RS, which randomly samples from the search space and possibly figures out more promising solutions.

Table 1: The average test accuracy of MF-NAS variants using random search (RS) and local search (LS) in the first stage. We experiment with the three worst-performance ZC proxies (i.e., Plain, Fisher, Zen) and the three best-performance ZC proxies (i.e., Synflow, FLOPs, Params). **Bold** results indicate the better method between RS and LS when using the same metric.

| Dataset | Plain | | Fisher | | Zen | | Synflow | | FLOPs | | Params | |
|---|---|---|---|---|---|---|---|---|---|---|---|---|
| | RS | LS | RS | LS | RS | LS | RS | LS | RS | LS | RS | LS |
| CIFAR-10 | **92.45** | 92.11 | **93.20** | 90.98 | **88.77** | 88.35 | 94.01 | **94.36** | 93.99 | **94.36** | 93.98 | **94.36** |
| CIFAR-100 | **69.64** | 68.93 | **70.07** | 65.67 | 60.24 | **60.70** | 72.17 | **73.51** | 72.52 | **73.51** | 72.53 | **73.51** |
| ImageNet16-120 | **43.29** | 42.74 | **44.79** | 35.97 | 35.66 | **40.77** | 46.24 | **46.34** | 45.93 | **46.34** | 45.89 | **46.34** |

### 5.1.3 COMPARISONS TO OTHER NAS METHODS

Table 2: Test accuracies (mean $\pm$ std) of best architectures found by algorithms on NAS-Bench-201. We only report the performance of the best three MF-NAS variants. The results of the remaining variants can be found in Table 6. The highest accuracies are presented in **bold**.

| Algorithm | Metrics | CIFAR-10 | CIFAR-100 | ImageNet16-120 |
|---|---|---|---|---|
| Random search | Validation accuracy | $93.35 \pm 0.66$ | $70.93 \pm 1.18$ | $44.82 \pm 1.23$ |
| Local search | Validation accuracy | $93.64 \pm 0.52$ | $71.43 \pm 0.84$ | $44.98 \pm 0.61$ |
| SH (Jamieson & Talwalkar, 2016) | Validation accuracy | $93.06 \pm 0.74$ | $70.36 \pm 1.18$ | $43.96 \pm 1.50$ |
| REA (Real et al., 2019) | Validation accuracy | $93.45 \pm 0.68$ | $71.20 \pm 1.16$ | $45.20 \pm 0.98$ |
| REA+W (Abdelfattah et al., 2021) | Synflow & Validation accuracy | $93.86 \pm 0.36$ | $71.69 \pm 0.73$ | $45.74 \pm 0.71$ |
| FreeREA (Cavagnero et al., 2023) | logSynflow + NWOT + Skip | $94.27 \pm 0.20$ | $73.10 \pm 0.49$ | $46.18 \pm 0.32$ |
| MF-NAS (Synflow) | Synflow & Validation accuracy | **94.36** $\pm$ **0.00** | **73.51** $\pm$ **0.05** | **46.34** $\pm$ **0.00** |
| MF-NAS (FLOPS) | FLOPS & Validation accuracy | **94.36** $\pm$ **0.00** | **73.51** $\pm$ **0.00** | **46.34** $\pm$ **0.00** |
| MF-NAS (Params) | Params & Validation accuracy | **94.36** $\pm$ **0.00** | **73.51** $\pm$ **0.00** | **46.34** $\pm$ **0.00** |
| Optimal (in the benchmark) | - | *94.37* | *73.51* | *47.31* |

Table 2 shows the superiority of MF-NAS to other *non-weight sharing* approaches. MF-NAS variants also exhibit an impressive stability since their variances are extremely small. Besides, the results indicate that MF-NAS is significantly more effective than running solely local search or Successive

Halving. MF-NAS also outperforms REA+W, which is an NAS algorithm using simultaneously both a ZC proxy and the validation accuracy metric. All results of the competing algorithms in Table 2 are reproduced by ourselves in our experiments. We also compare our MF-NAS to other state-of-the-art non-weight sharing methods in Appendix A.4 and find that ours is the best among all methods.

## 5.2 RESULTS ON OTHER SEARCH SPACES

We evaluate our methods on NAS-Bench-101 (Ying et al., 2019) and NAS-Bench-ASR (Mehrotra et al., 2021) search spaces. Table 3 exhibits that when searching under the same budgets (i.e., 20,000 seconds for NAS-Bench-101 and 300 training epochs for NAS-Bench-ASR), there is no approach that obtains comparable performance to our MF-NAS variants. We also compare MF-NAS to other state-of-the-art methods and find that there is no method that uses less computational costs but performs better than ours (see Appendix A.4).

Table 3: Test accuracies and test phoneme error rates (PER) (mean $\pm$ std) of best architectures found by algorithms on NAS-Bench-101 and NAS-Bench-ASR, respectively. The highest accuracy and the lowest PER are presented in **bold**.

| Algorithm | Metrics | NAS-Bench-101 | NAS-Bench-ASR |
|---|---|---|---|
| Random search | Validation accuracy | $93.16 \pm 0.26$ | $22.15 \pm 0.45$ |
| Local search | Validation accuracy | $93.16 \pm 0.56$ | $22.51 \pm 2.85$ |
| SH (Jamieson & Talwalkar, 2016) | Validation accuracy | $93.19 \pm 0.46$ | $22.29 \pm 0.60$ |
| REA (Real et al., 2019) | Validation accuracy | $93.24 \pm 0.27$ | $22.32 \pm 0.71$ |
| REA+W (Abdelfattah et al., 2021) | Synflow + Validation accuracy | $93.22 \pm 0.28$ | $22.02 \pm 0.25$ |
| FreeREA (Cavagnero et al., 2023) | logSynflow + NWOT + Skip | $93.53 \pm 0.70$ | - |
| MF-NAS (Synflow) | Synflow & Validation accuracy | $93.82 \pm 0.56$ | $\mathbf{21.77 \pm 0.00}$ |
| MF-NAS (FLOPS) | FLOPS & Validation accuracy | $93.88 \pm 0.25$ | $21.80 \pm 0.37$ |
| MF-NAS (Params) | Params & Validation accuracy | $\mathbf{93.89 \pm 0.25}$ | $21.81 \pm 0.26$ |
| Optimal (in the benchmark) | - | *94.37* | *21.40* |

## 6 CONCLUSION

In this study, we investigated the effectiveness of zero-cost (ZC) proxies in guiding a typical local search algorithm to seek for top-performing neural network architectures. Fitness landscape analyses for a series of ZC proxies showed that using solely ZC proxies was not effective and the obtained architectures were inferior compared to using training-based metrics such as validation accuracy or training loss. The accuracy values of architectures at the global optima of ZC proxy landscapes are much lower than the optimal accuracy, and there are few architectures that simultaneously have high ZC proxy scores and high test accuracies. It explains why NAS algorithms using a ZC proxy as the search objective might return subpar architectures although they figure out the networks that have high ZC proxy scores. Next, we observed that the landscapes of ZC proxies allowed local search algorithms to approach easily and efficiently the regions containing top-performing architectures although the landscapes are multi-modal. Based on these insights, we introduced the novel Multi-Fidelity Neural Architecture Search (MF-NAS) framework, which makes use of both ZC metrics and training-based metrics to find high-performance architectures with an acceptable cost. MF-NAS first explores the search space using a ZC-proxy-guided local search and the architectures that have the highest ZC proxy scores are then forwarded to the Successive Halving procedure that uses a training-based metric to yield a final best architecture with its associated network weights. Extensive experiments on NAS-Bench-101, NAS-Bench-201, and NAS-Bench-ASR benchmarks demonstrated the consistent efficacy and efficiency of our MF-NAS in achieving high-quality architectures compared to all state-of-the-art training-free NAS algorithms.

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

# A APPENDIX

## A.1 ABLATION RESULTS

### A.1.1 FULL ANALYSIS RESULTS ON NAS-BENCH-201

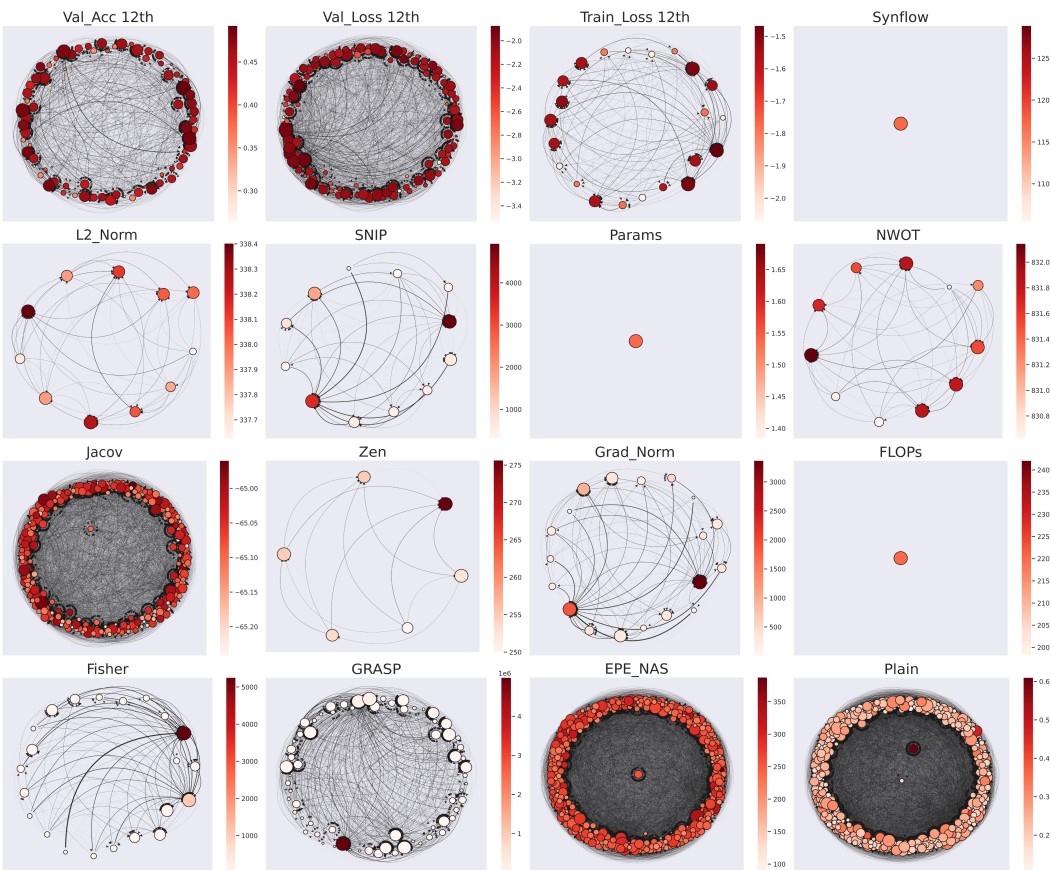

Figure 6: MLONs of the best-improvement hill climbing algorithm with escape for CIFAR-100 on NAS-Bench-201.

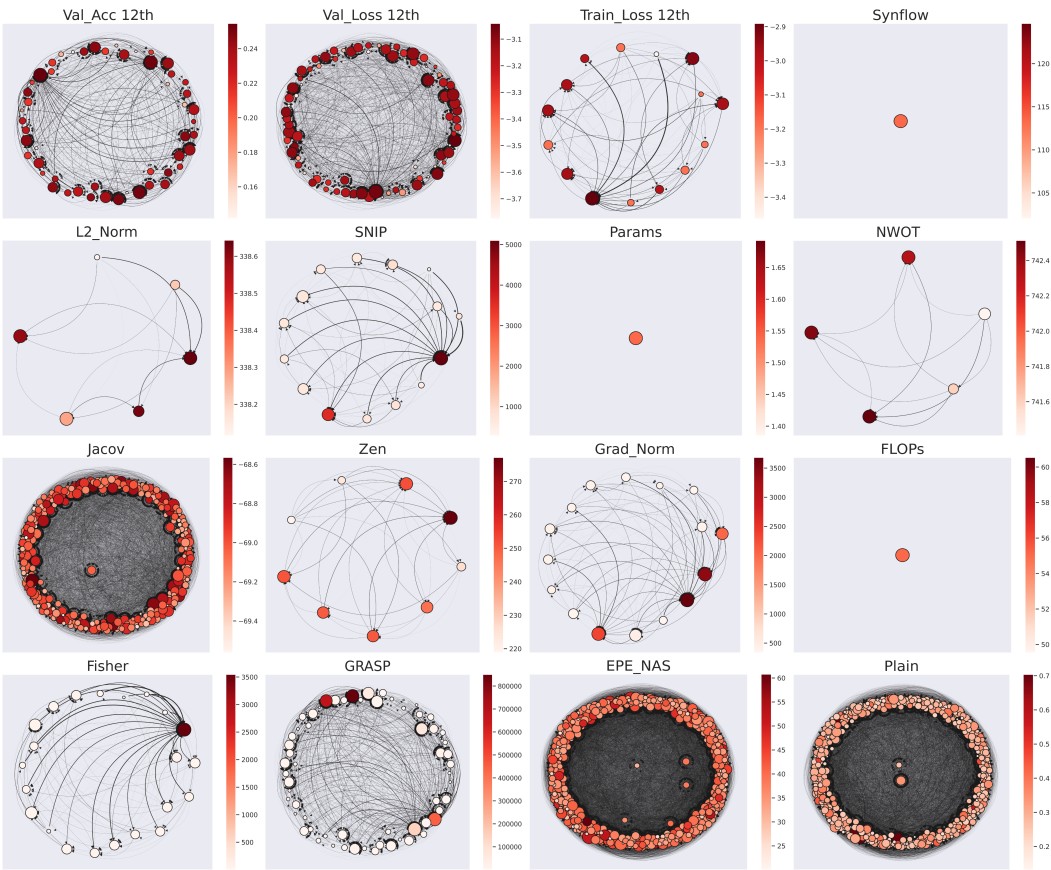

Figure 7: MLONs of the best-improvement hill climbing algorithm with escape for ImageNet16-120 on NAS-Bench-201.

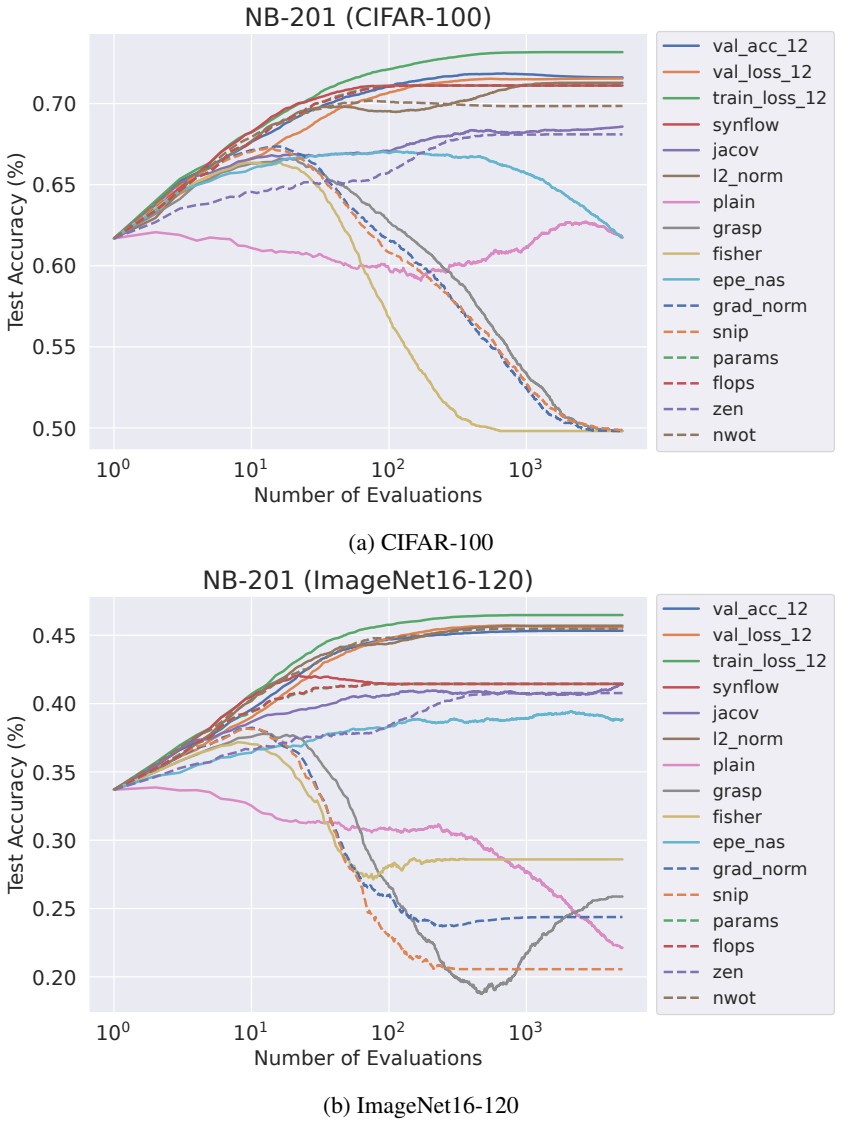

(a) CIFAR-100

(b) ImageNet16-120

Figure 8: The average performance trends of the first-improvement hill climbing algorithm with escape regarding 3 training-based metrics and 13 ZC proxies over 5,000 evaluations on (a) CIFAR-100 and (b) ImageNet16-120. We repeat the search process 500 times for each metric.

### A.1.2 FULL RESULTS OF MF-NAS VARIANTS

Table 4: The average test accuracy $(\cdot/)$ of the architectures chosen by Successive Halving (SH) and the true best architectures in the top-32 candidate architectures over 500 runs of MF-NAS on NAS-Bench-201. For the results of SH, we also report the average rank $(/\cdot)$ of the selected architecture out of 32 candidates (rank 1 is the best).

| **Metric** | **CIFAR-10** | | **CIFAR-100** | | **ImageNet16-120** | |
|---|---|---|---|---|---|---|
| | Top-32 | SH | Top-32 | SH | Top-32 | SH |
| Jacov | 93.82 | 93.63 / 2.2 | 71.91 | 71.73 / 1.9 | 45.33 | 44.40 / 4.3 |
| Plain | 92.49 | 92.11 / 2.3 | 69.14 | 68.93 / 1.5 | 43.40 | 42.74 / 1.9 |
| Grasp | 93.11 | 92.77 / 2.4 | 69.47 | 69.42 / 1.2 | 42.87 | 42.77 / 1.2 |
| Fisher | 92.35 | 90.98 / 15.3 | 66.72 | 65.67 / 6.0 | 38.83 | 35.97 / 2.9 |
| EPE-NAS | 93.89 | 93.40 / 5.7 | 71.76 | 71.51 / 2.1 | 44.94 | 44.12 / 3.1 |
| Grad-norm | 93.13 | 92.58 / 2.1 | 68.85 | 68.82 / 1.1 | 38.83 | 35.97 / 3.1 |
| Snip | 93.13 | 93.13 / 1.0 | 69.33 | 69.28 / 1.1 | 38.83 | 35.97 / 3.5 |
| Synflow | 94.37 | 94.36 / 2.0 | 73.51 | 73.51 / 1.0 | 46.48 | 46.34 / 2.6 |
| L2-norm | 93.76 | 93.32 / 14.8 | 71.59 | 70.80 / 10.4 | 46.53 | 46.48 / 1.3 |
| Zen | 90.74 | 88.35 / 20.8 | 68.55 | 60.70 / 21.7 | 40.77 | 40.77 / 1.0 |
| NWOT | 93.89 | 93.33 / 14.1 | 71.74 | 71.16 / 5.2 | 46.55 | 45.32 / 13.0 |
| Params | 94.36 | 94.36 / 1.0 | 73.51 | 73.51 / 1.0 | 46.34 | 46.34 / 1.0 |
| FLOPs | 94.36 | 94.36 / 1.0 | 73.51 | 73.51 / 1.0 | 46.34 | 46.34 / 1.0 |

Table 5: Performance of MF-NAS variants using the validation accuracy and training loss in the SH procedure (both metrics are queried at the 12-th epoch). The enhanced performance obtained by replacing the validation accuracy with the training loss is presented in **bold**.

| Metric | CIFAR-10 | | CIFAR-100 | | ImageNet16-120 | |
|---|---|---|---|---|---|---|
| Metric | Val Acc | Train Loss | Val Acc | Train Loss | Val Acc | Train Loss |
| Jacov | $93.63 \pm 0.43$ | $\mathbf{93.78 \pm 0.23}$ | $71.73 \pm 0.67$ | $\mathbf{71.79 \pm 0.50}$ | $44.40 \pm 0.90$ | $\mathbf{44.47 \pm 1.01}$ |
| Plain | $92.11 \pm 0.96$ | $\mathbf{92.42 \pm 0.63}$ | $68.93 \pm 1.15$ | $68.61 \pm 1.18$ | $42.74 \pm 2.35$ | $\mathbf{42.83 \pm 2.04}$ |
| Grasp | $92.77 \pm 0.32$ | $\mathbf{93.09 \pm 0.12}$ | $69.42 \pm 0.17$ | $69.35 \pm 0.13$ | $42.77 \pm 1.42$ | $\mathbf{42.87 \pm 1.13}$ |
| Fisher | $90.98 \pm 1.07$ | $\mathbf{92.21 \pm 0.62}$ | $65.67 \pm 0.91$ | $\mathbf{65.72 \pm 0.73}$ | $35.97 \pm 0.00$ | $\mathbf{38.83 \pm 0.00}$ |
| EPE-NAS | $93.40 \pm 0.61$ | $\mathbf{93.98 \pm 0.29}$ | $71.51 \pm 1.14$ | $\mathbf{71.58 \pm 0.98}$ | $44.12 \pm 1.19$ | $\mathbf{44.34 \pm 1.02}$ |
| Grad-norm | $92.58 \pm 0.44$ | $\mathbf{93.13 \pm 0.06}$ | $68.82 \pm 0.77$ | $68.25 \pm 1.59$ | $35.97 \pm 0.00$ | $\mathbf{38.83 \pm 0.00}$ |
| Snip | $93.13 \pm 0.00$ | $93.13 \pm 0.00$ | $69.28 \pm 0.34$ | $\mathbf{69.33 \pm 0.00}$ | $35.97 \pm 0.00$ | $\mathbf{38.83 \pm 0.00}$ |
| Synflow | $94.36 \pm 0.00$ | $94.35 \pm 0.04$ | $73.51 \pm 0.05$ | $73.51 \pm 0.00$ | $46.34 \pm 0.00$ | $46.34 \pm 0.00$ |
| L2-norm | $93.32 \pm 0.03$ | $\mathbf{93.67 \pm 0.04}$ | $70.80 \pm 0.37$ | $\mathbf{71.15 \pm 0.12}$ | $46.48 \pm 0.21$ | $\mathbf{46.53 \pm 0.04}$ |
| Zen | $88.35 \pm 0.55$ | $\mathbf{90.64 \pm 0.00}$ | $60.70 \pm 0.76$ | $\mathbf{66.11 \pm 0.00}$ | $40.77 \pm 0.00$ | $40.32 \pm 0.00$ |
| NWOT | $93.35 \pm 0.07$ | $\mathbf{93.56 \pm 0.01}$ | $71.16 \pm 0.46$ | $\mathbf{71.28 \pm 0.25}$ | $45.32 \pm 0.06$ | $\mathbf{46.53 \pm 0.05}$ |
| Params | $94.36 \pm 0.00$ | $94.36 \pm 0.00$ | $73.51 \pm 0.00$ | $73.51 \pm 0.00$ | $46.34 \pm 0.00$ | $46.34 \pm 0.00$ |
| FLOPS | $94.36 \pm 0.00$ | $94.36 \pm 0.00$ | $73.51 \pm 0.00$ | $73.51 \pm 0.00$ | $46.34 \pm 0.00$ | $46.34 \pm 0.00$ |

Table 6: Performance of MF-NAS variants with different zero-cost metrics on NAS-Bench-201.

| Metric | CIFAR-10 | CIFAR-100 | ImageNet16-120 |
|---|---|---|---|
| Jacov | $93.63 \pm 0.43$ | $71.73 \pm 0.67$ | $44.40 \pm 0.90$ |
| Plain | $92.11 \pm 0.96$ | $68.93 \pm 1.15$ | $42.74 \pm 2.35$ |
| Grasp | $92.77 \pm 0.32$ | $69.42 \pm 0.17$ | $42.77 \pm 1.42$ |
| Fisher | $90.98 \pm 1.07$ | $65.67 \pm 0.91$ | $35.97 \pm 0.00$ |
| EPE-NAS | $93.40 \pm 0.61$ | $71.51 \pm 1.14$ | $44.12 \pm 1.19$ |
| Grad-norm | $92.58 \pm 0.44$ | $68.82 \pm 0.77$ | $35.97 \pm 0.00$ |
| Snip | $93.13 \pm 0.00$ | $69.28 \pm 0.34$ | $35.97 \pm 0.00$ |
| Synflow | $94.36 \pm 0.00$ | $73.51 \pm 0.05$ | $46.34 \pm 0.00$ |
| L2-norm | $93.32 \pm 0.03$ | $70.80 \pm 0.37$ | $46.48 \pm 0.21$ |
| Zen | $88.35 \pm 0.55$ | $60.70 \pm 0.76$ | $40.77 \pm 0.00$ |
| NWOT | $93.35 \pm 0.07$ | $71.16 \pm 0.46$ | $45.32 \pm 0.06$ |
| Params | $94.36 \pm 0.00$ | $73.51 \pm 0.00$ | $46.34 \pm 0.00$ |
| FLOPS | $94.36 \pm 0.00$ | $73.51 \pm 0.00$ | $46.34 \pm 0.00$ |

A.2    PSEUDOCODE OF MULTI-FIDELITY NEURAL ARCHITECTURE SEARCH FRAMEWORK

---

**Algorithm 1:** Multi-Fidelity Local Search Successive Halving (MF-NAS)

---

**Input:** Zero-Cost proxy $Z$
Training-based metric $T$
Maximum number of evaluations in the first stage $M$
The number of top architectures for selection to the second stage $k$
Total budget $B$ (in seconds)
**Output:** The best architecture found so far $\boldsymbol{x}^*$.

1  $\mathcal{H}, B \leftarrow$ Zero-Cost-Local-Search$(Z, M, B)$        // First stage:  Exploring search space using First-improvement Hill Climbing with zero-cost metric $Z$.

2  $P \leftarrow$ Get-TopK-Solutions$(\mathcal{H}, k)$

3  $\boldsymbol{x}^* \leftarrow$ Successive-Halving$(P, T, B)$        // Second stage:  Selecting the best architecture $\boldsymbol{x}^*$ from the candidate pool $P$ using Successive Halving with the training-based metric $T$.

4  **return** $\boldsymbol{x}^*$

---

## A.3 EXPERIMENTAL DETAILS

**Random search (SH)** In our experiment, the RS iterates the process of randomly sampling the architectures from the search space and evaluating their validation accuracy (or PER for NAS-Bench-ASR) at the $12^{th}$ epoch till the allowed budget runs out. The architecture that has the highest accuracy so far is the one returned by RS when the search finishes.

**Local search (LS)** The local search algorithm implemented in our experiments is the first-improvement hill-climbing algorithm that we employ in the first stage of MF-NAS but use the validation accuracy (or PER for NAS-Bench-ASR) at the $12^{th}$ epoch as the search objective instead of a ZC proxy. Similar to RS, the procedure of LS is performed till the allowed budget is exhausted and the architecture that has the highest accuracy so far is considered the resulting solution found by LS.

**Successive Halving (SH)** We refer to Jamieson & Talwalkar (2016) to implement the SH method in our experiments. For the variant that solely uses SH, the input architectures are randomly sampled from the search space while the input candidates for SH in our MF-NAS framework are obtained via the ZC proxy-guided local search. The number of input architectures $k$ is set to 32 (for NAS-Bench-201) and 16 (for NAS-Bench-101 and NAS-Bench-ASR). For NAS-Bench-201, the architectures are evaluated for their validation accuracies at 5 different epochs: $12^{th}$, $25^{th}$, $50^{th}$, $100^{th}$, and $200^{th}$. For NAS-Bench-101 and NAS-Bench-ASR, we evaluate architectures at 4 different epoch: $4^{th}$, $12^{th}$, $36^{th}$, and $108^{th}$ for NAS-Bench-101, and $10^{th}$, $20^{th}$, $30^{th}$ and $40^{th}$ for NAS-Bench-ASR.

**Regularized Evolution (REA)** REA (Real et al., 2019) is an evolution-based algorithm in which individuals in the population represent architectures. In the beginning, REA randomly samples $N$ architectures (in which $N$ is the population size) from the search space to form the initial population. The individual that has the highest fitness (e.g., the highest accuracy (or PER for NAS-Bench-ASR) at the $12^t h$ epoch) in the population is selected by using the tournament selection with the tournament size $s$. The selected individual is then used to produce the new offspring via the mutation operator with probability $p_M$. The newly-created architecture is added to the population and the oldest individual is eliminated from the population. REA iterates the process of producing and eliminating architectures till the stopping condition is satisfied (e.g., reaching the maximum search time). The individual that has the highest fitness so far during the search is considered as the optimal architecture found by REA. In our experiments, the hyperparameters of REA are set as in Dong et al. (2022); Dong & Yang (2020), i.e., population size $N$ and tournament size $s$ are 10, the mutation probability $p_M$ is $1/l$ in which $l$ is the length of encoded architecture.

**Regularized Evolution + Warmup (REA+W)** REA+W uses the Warmup procedure (Abdelfattah et al., 2021) to create the initial population. Specifically, REA+W randomly samples a large number of architectures from the search space and uses the Synaptic Flow metric (Synflow) to evaluate sampled architectures for efficiency. The top-$N$ (in which $N$ is the population size of REA) networks that have the highest Synflow scores are then selected as the individuals of the initial population of REA. The remaining parts of REA+W are executed as in REA. We set the number of sampled architectures in the Warmup stage to 2,000 for all cases.

**FreeREA** (Cavagnero et al., 2023) is a training-free evolution-based method that combines three zero-cost metrics into a single search objective to guide the Regularized Evolution algorithm. We refer the readers to Cavagnero et al. (2023) for more details on this method. The hyperparameters of FreeREA are set at their default values in our experiments.

## A.4 COMPARISONS TO STATE-OF-THE-ART NAS METHODS

We further compare our MF-NAS to state-of-the-art *non-weight-sharing* and *weight-sharing* NAS methods on NAS-Bench-101 and NAS-Bench-201 search spaces.

For NAS-Bench-201, Table 7 indicates that MF-NAS variants are the best at all three datasets regarding the test accuracies of the obtained architectures. In terms of the search cost, our methods obviously have a higher cost compared to entirely training-free methods such as NASWOT, EPE-NAS, or FreeREA. However, we note that the cost of our methods has included the time of fully training the returned architecture (during the Successive Halving). That means when the search is finished, the networks returned by MF-NAS are fully-trained and their network weights can be straightforwardly employed. In contrast, training-free methods only return the topology of the networks and thus, a full training process is required to obtain the proper network weights.

Table 7: Test accuracy comparisons with other *non-weight-sharing* methods on NAS-Bench-201. The results of competitive methods are taken from Yan et al. (2020); Mellor et al. (2021); Chen et al. (2021b); Lopes et al. (2021; 2022); Dong et al. (2022); Yang et al. (2023). The first and second blocks represent the algorithms using training-based metrics and training-free metrics as search objective, respectively. The highest accuracies are presented in **bold**.

| Algorithm | CIFAR-10 | | CIFAR-100 | | ImageNet16-120 | |
|---|---|---|---|---|---|---|
| | Search Cost (seconds) | Accuracy (%) | Search Cost (seconds) | Accuracy (%) | Search Cost (seconds) | Accuracy (%) |
| Random Search | 20,000 | $93.73 \pm 0.36$ | 40,000 | $71.62 \pm 0.91$ | 120,000 | $45.30 \pm 1.00$ |
| Local Search | 20,000 | $94.03 \pm 0.38$ | 40,000 | $72.35 \pm 0.93$ | 120,000 | $45.94 \pm 0.73$ |
| SH (Jamieson & Talwalkar, 2016) | 20,000 | $93.22 \pm 0.66$ | 40,000 | $70.07 \pm 1.41$ | 120,000 | $43.59 \pm 1.80$ |
| REA (Real et al., 2019) | 20,000 | $93.85 \pm 0.40$ | 40,000 | $72.32 \pm 0.81$ | 120,000 | $45.72 \pm 0.79$ |
| REA+W (Abdelfattah et al., 2021) | 20,000 | $94.21 \pm 0.20$ | 40,000 | $72.75 \pm 0.52$ | 120,000 | $46.21 \pm 0.52$ |
| REINFORCE (Williams, 1992) | 20,000 | $93.85 \pm 0.40$ | 40,000 | $72.32 \pm 0.81$ | 120,000 | $45.72 \pm 0.79$ |
| BOHB (Falkner et al., 2018) | 20,000 | $93.85 \pm 0.40$ | 40,000 | $72.32 \pm 0.81$ | 120,000 | $45.72 \pm 0.79$ |
| arch2vec-RL (Yan et al., 2020) | 12,000 | $94.12 \pm 0.42$ | - | $73.15 \pm 0.78$ | - | $46.16 \pm 0.38$ |
| arch2vec-BO (Yan et al., 2020) | 12,000 | $94.18 \pm 0.24$ | - | $73.37 \pm 0.30$ | - | $46.27 \pm 0.37$ |
| G-EA (Lopes et al., 2022) | 26,911 | $93.99 \pm 0.23$ | - | $72.36 \pm 0.66$ | - | $46.04 \pm 0.67$ |
| PRE-NAS (Peng et al., 2022) | 20,000 | $94.04 \pm 0.34$ | 40,000 | $72.02 \pm 1.22$ | 120,000 | $45.34 \pm 1.03$ |
| NASWOT (Mellor et al., 2021) | 307 | $92.96 \pm 0.81$ | 307 | $69.98 \pm 1.22$ | 307 | $44.44 \pm 2.10$ |
| EPE-NAS (Lopes et al., 2021) | 207 | $91.31 \pm 1.69$ | 207 | $69.58 \pm 0.83$ | 207 | $41.84 \pm 2.06$ |
| TE-NAS (Chen et al., 2021b) | 1,558 | $93.90 \pm 0.47$ | - | $71.24 \pm 0.56$ | - | $42.38 \pm 0.46$ |
| ST-NAS (Yang et al., 2023) | 437 | $93.46 \pm 0.59$ | - | $70.58 \pm 0.82$ | - | $43.74 \pm 1.48$ |
| FreeREA (Cavagnero et al., 2023) | 45 | $94.27 \pm 0.20$ | 45 | $73.10 \pm 0.49$ | 45 | $46.18 \pm 0.32$ |
| MF-NAS (Synflow) | 20,000 | $\mathbf{94.36 \pm 0.00}$ | 40,000 | $\mathbf{73.51 \pm 0.05}$ | 120,000 | $\mathbf{46.34 \pm 0.00}$ |
| MF-NAS (FLOPS) | 20,000 | $\mathbf{94.36 \pm 0.00}$ | 40,000 | $\mathbf{73.51 \pm 0.00}$ | 120,000 | $\mathbf{46.34 \pm 0.00}$ |
| MF-NAS (Params) | 20,000 | $\mathbf{94.36 \pm 0.00}$ | 40,000 | $\mathbf{73.51 \pm 0.00}$ | 120,000 | $\mathbf{46.34 \pm 0.00}$ |
| Optimal (in the benchmark) | - | *94.37* | - | *73.51* | - | *47.31* |

Compared to weight-sharing NAS methods, MF-NAS variants are only worse than Shapley-NAS (Xiao et al., 2022) on CIFAR-10 (94.36% compared to 94.37%) and ImageNet16-120 (46.34% compared to 46.85%) (see Table 8). However, similar to training-free NAS approaches, the main drawback of weight-sharing methods is that the resulting architectures need to be trained from scratch, or at least fine-tuning, to achieve their optimal weight values. We also emphasize that the computation costs of weight-sharing methods are often much higher than non-weight-sharing methods like ours because they need to pretrain a supernet and maintain it during the search.

In the case of NAS-Bench-101, Table 9 demonstrates that all MF-NAS variants significantly outperform weight-sharing methods and perform better than nearly all non-weight-sharing methods (except for RankNOSH (Wang et al., 2021)). However, it is worth noting that RankNOSH incurs a considerably higher search cost, requiring 8,400 training epochs, which is over 20 times greater than ours (i.e., 368 epochs). We further re-run MF-NAS but adjust the checkpoints for evaluating architectures in the Successive Halving procedure (denoted as MF-NAS[†] in Table 9). Remarkably, MF-NAS[†] surpasses RankNOSH with a search cost that is 7 times less.

Table 8: Test accuracy comparisons with *weight-sharing* methods on NAS-Bench-201. The results of competitive methods are taken from Dong et al. (2022); Xiao et al. (2022). The highest accuracies are presented in **bold**.

| Algorithm | CIFAR-10 | CIFAR-100 | ImageNet16-120 |
|---|---|---|---|
| ENAS (Pham et al., 2018) | $93.76 \pm 0.00$ | $70.67 \pm 0.62$ | $41.44 \pm 0.00$ |
| RSPS (Li & Talwalkar, 2019) | $91.05 \pm 0.66$ | $68.26 \pm 0.96$ | $40.69 \pm 0.36$ |
| DARTS (1st) (Liu et al., 2019) | $59.84 \pm 7.84$ | $61.26 \pm 4.43$ | $37.88 \pm 2.91$ |
| DARTS (2nd) (Liu et al., 2019) | $65.38 \pm 7.84$ | $60.49 \pm 4.95$ | $36.79 \pm 7.59$ |
| GDAS (Dong & Yang, 2019a) | $93.23 \pm 0.58$ | $68.17 \pm 2.50$ | $39.40 \pm 0.00$ |
| SETN (Dong & Yang, 2019b) | $92.72 \pm 0.73$ | $69.36 \pm 1.72$ | $39.51 \pm 0.33$ |
| PC-DARTS (Xu et al., 2020) | $93.76 \pm 0.00$ | $70.67 \pm 0.62$ | $41.44 \pm 0.00$ |
| DrNAS (Chen et al., 2021c) | $94.36 \pm 0.00$ | $\mathbf{73.51 \pm 0.00}$ | $46.34 \pm 0.00$ |
| ShapleyNAS (Xiao et al., 2022) | $\mathbf{94.37 \pm 0.00}$ | $\mathbf{73.51 \pm 0.00}$ | $\mathbf{46.85 \pm 0.12}$ |
| MF-NAS (Synflow) | $94.36 \pm 0.00$ | $\mathbf{73.51 \pm 0.05}$ | $46.34 \pm 0.00$ |
| MF-NAS (FLOPS) | $94.36 \pm 0.00$ | $\mathbf{73.51 \pm 0.00}$ | $46.34 \pm 0.00$ |
| MF-NAS (Params) | $94.36 \pm 0.00$ | $\mathbf{73.51 \pm 0.00}$ | $46.34 \pm 0.00$ |
| Optimal (in the benchmark) | *94.37* | *73.51* | *47.31* |

Table 9: Test accuracy comparisons with state-of-the-art methods on NAS-Bench-101. The results of state-of-the-art methods are taken from Yu et al. (2020); Mellor et al. (2021); Wang et al. (2021). The highest accuracies are presented in **bold**.

| **Weight-sharing** | | |
|---|---|---|
| **Algorithm** | Accuracy (%) | Search Cost (seconds) |
| ENAS (Pham et al., 2018) | $91.83 \pm 0.42$ | - |
| NAO (Luo et al., 2018) | $92.59 \pm 0.59$ | - |
| FBNet (Wu et al., 2019) | $92.29 \pm 1.25$ | - |
| DARTS (Liu et al., 2019) | $92.21 \pm 0.61$ | - |
| SPOS (Guo et al., 2020) | $89.85 \pm 3.80$ | - |
| FairNAS (Chu et al., 2021) | $91.10 \pm 1.84$ | - |
| **Non weight-sharing** | | |
| Random search | $93.16 \pm 0.26$ | 20,000 |
| Local search | $93.16 \pm 0.56$ | 20,000 |
| SH (Jamieson & Talwalkar, 2016) | $93.19 \pm 0.46$ | 20,000 |
| REA (Real et al., 2019) | $93.24 \pm 0.27$ | 20,000 |
| REA+W (Abdelfattah et al., 2021) | $93.22 \pm 0.28$ | 20,000 |
| RankNOSH (Wang et al., 2021) | 93.97 | 8,400 epochs |
| NASWOT (Mellor et al., 2021) | $91.77 \pm 0.05$ | 23 |
| FreeREA (Cavagnero et al., 2023) | $93.53 \pm 0.70$ | 750 |
| MF-NAS (Synflow) | $93.82 \pm 0.56$ | 14,742 (= 368 epochs) |
| MF-NAS (FLOPS) | $93.88 \pm 0.25$ | 13,487 (= 368 epochs) |
| MF-NAS (Params) | $93.89 \pm 0.25$ | 13,517 (= 368 epochs) |
| MF-NAS[†] (Params) | $\mathbf{94.07 \pm 0.20}$ | 42,924 (= 1,152 epochs) |
| Optimal (in the benchmark) | *94.37* | - |

