# OpenReview forum: "Efficient Multi-Fidelity NAS with Zero-Cost Proxy-Guided Local Search"
_ICLR.cc/2024/Conference — ICLR 2024 Conference Withdrawn Submission_

### Official Review · Reviewer_K4zY · 2023-11-02

**Soundness:** 2 fair
**Presentation:** 3 good
**Contribution:** 2 fair
**Rating:** 3
**Confidence:** 4

**Summary:**

This paper exploits the capabilities of zero-cost proxies in NAS to identify good regions of the search space and later on employs a multi-fidelity method (SuccessiveHalving) in order to optimize in this region identified by local search using the ZCPs as objective function. The authors evaluate their algorithm on standard tabular NAS benchmarks such as NB101, NB201 and NB-ASR, achieving better performance compared to other blackbox NAS optimizers.

**Strengths:**

- The paper provides some interesting insights on the performance of ZCPs on standard NAS tabular benchmarks. The visualization via MLON is interesting and backs up the reasoning behind using local search to identify promising regions in a multimodal landscape defined by the proxies' objective.

- In general I like the idea to identify promising regions of large search spaces in order to exploit more there by allocating more resources. The idea of using SuccessiveHalving (SH) is simple and provides an effective way to find stress more exploitation in the end.

- The results look promising in the benchmarks that the authors picked.

- Easy to follow and the visualizations are clear and informative enough to quickly get the gist of the paper.

- Available code.

**Weaknesses:**

- **Novelty**: The proposed algorithm combines existing methods for NAS and multi-fidelity optimization, therefore I do not see much contributions in methodological novelty. Granted, the authors determine to use Local Search instead of Random Search for unimodal ZCP distributions, based on their loss landscape observations, however I am not sure how effective this is for ZCP on larger search spaces.

- **More benchmarks**: The authors did evaluate their algorithm only on 3 tabular benchmarks, while there are many more available in NAS-Bench-Suite-Zero. Moreover, I think it is still important to benchmark the performance of MF-NAS in other standard benchmarks (at least even CIFAR-10, DARTS space).

- **Comparison to one-shot NAS**: In the appendix the authors compare to one-shot NAS approaches, such as ShapleyNAS and DrNAS, which achieve the same performance in NB201 as their method. ShapleyNAS and DrNAS are already quite efficient algorithms, so this puzzles me a bit regarding the usefulness of MF-NAS.

**Questions:**

- Is there any reason why the authors didn't evaluate their approach on all the benchmarks in NAS-Bench-Suite-Zero?

- Can the authors say a few more words regarding the comparison to one-shot NAS methods, especially related to the search costs?

---

### Official Review · Reviewer_eGD3 · 2023-11-04

**Soundness:** 2 fair
**Presentation:** 3 good
**Contribution:** 2 fair
**Rating:** 3
**Confidence:** 5

**Summary:**

This paper raises a good question for NAS researchers: whether ZC metrics are essential and old-fashion training-based metrics may be better. The paper investigates the effectiveness of ZC proxies in NAS by taking a deeper look into their fitness landscapes rather than focusing only on rank correlation. Furthermore, the authors proposed a method combining ZC metrics and training-based metrics. The experiments show the good performance of the proposed approach.

**Strengths:**

1. The studied problem is important. ZC metrics are widely utilized in automated machine learning for finding good architectures.
2. The paper is written well.
3. The idea of combining ZC metrics and training-based metrics makes sense.

**Weaknesses:**

1. The experiments are weak. In experiments, the authors have missed those recent advances of utilizing ZC metrics in neural architecture search, some of which are listed as follows:

- Proxybo: Accelerating neural architecture search via bayesian optimization with zero-cost proxies, AAAI 2022
- RD-NAS: Enhancing One-shot Supernet Ranking Ability via Ranking Distillation from Zero-cost Proxies, ICASSP 2023.
- Zero-cost operation scoring in differentiable architecture search, AAAI 2023
- Extensible and Efficient Proxy for Neural Architecture Search, ICCV 2023.

That is, both the preliminary analysis and performance comparison lack of the study of the recent methods of this area.

2. To claim the limitation of ZC metrics, more theoretical analysis is required.

**Questions:**

My questions include two parts, corresponding to the Weaknesses.

(1) Why don't you consider the recent advances in this area?
(2) What about the theoretical basis?

---

### Official Review · Reviewer_Kzxp · 2023-11-06

**Soundness:** 2 fair
**Presentation:** 3 good
**Contribution:** 2 fair
**Rating:** 3
**Confidence:** 4

**Summary:**

This work constructs fitness landscapes for several ZC proxy-based local search through LON visualization tool. The authors conclude that a high correlation does not guarantee finding high-performance architecture, and ZC proxies with low correlation could still be better in certain situations. Therefore, the authors introduce a two-stage NAS method, named MF-NAS by first shrinking the search space through ZC proxy-based local search and then adopt successive halving strategy to search on the small space.

**Strengths:**

1.	It is quite interesting to construct fitness landscapes through LON visualization tool.

**Weaknesses:**

1.	I am afraid that the visualization results are not so convincing. On the one hand, the search space is quite small (NAS-Bench-201); On the other hand, only local search algorithm is explored, the results and conclusion may not suitable for EA or BO.

2.	The novelty is limited. The local search and successive halving algorithm is widely used in NAS tasks.

**Questions:**

Please see the weaknesses.